# Suspected Malignant Thyroid Nodules in Children and Adolescents According to Ultrasound Elastography and Ultrasound-Based Risk Stratification Systems—Experience from One Center

**DOI:** 10.3390/jcm11071768

**Published:** 2022-03-23

**Authors:** Hanna Borysewicz-Sańczyk, Beata Sawicka, Agata Karny, Filip Bossowski, Katarzyna Marcinkiewicz, Aleksandra Rusak, Janusz Dzięcioł, Artur Bossowski

**Affiliations:** 1Department of Pediatrics, Endocrinology, Diabetology with Cardiology Unit, Medical University of Bialystok, 15-089 Bialystok, Poland; beata.sawicka@umb.edu.pl (B.S.); abossowski@hotmail.com (A.B.); 2Student Research Group by the Department of Pediatrics, Endocrinology, Diabetology with Cardiology Unit, Medical University of Bialystok, 15-089 Bialystok, Poland; agatakarny@gmail.com (A.K.); bossowski.filip@gmail.com (F.B.); katarzynamarcinkiewicz95@gmail.com (K.M.); olarusak008@gmail.com (A.R.); 3Department of Human Anatomy, Medical University of Bialystok, 15-089 Bialystok, Poland; janusz.dzieciol@umb.edu.pl

**Keywords:** thyroid nodules, thyroid cancer, children, thyroid ultrasonography, elastography, BTA, ATA, ultrasound risk-classification system

## Abstract

The risk of malignancy in thyroid nodules correlates with the presence of ultrasonographic features. In adults, ultrasound risk-classification systems have been proposed to indicate the need for further invasive diagnosis. Furthermore, elastography has been shown to support differential diagnosis of thyroid nodules. The purpose of our study was to assess the application of the American Thyroid Association (ATA), British Thyroid Association (BTA) ultrasound risk-classification systems and strain elastography in the management of thyroid nodules in children and adolescents from one center. Seventeen nodules with Bethesda III, IV, V and VI were selected from 165 focal lesions in children. All patients underwent ultrasonography and elastography followed by fine needle aspiration biopsy. Ultrasonographic features according to the ATA and BTA stratification systems were assessed retrospectively. The strain ratio in the group of thyroid nodules diagnosed as malignant was significantly higher than in benign nodules (6.07 vs. 3.09, *p* = 0.036). According to the ATA guidelines, 100% of malignant nodules were classified as high suspicion and 73% of benign nodules were assessed as low suspicion. Using the BTA U-score classification, 80% of malignant nodules were classified as cancerous (U5) and 20% as suspicious for malignancy (U4). Among benign nodules, 82% were classified as indeterminate or equivocal (U3) and 9% as benign (U2). Our results suggest that application of the ATA or BTA stratification system and elastography may be a suitable method for assessing the level of suspected malignancy in thyroid nodules in children and help make a clinical decision about the need for further invasive diagnosis of thyroid nodules in children.

## 1. Introduction

Thyroid cancer (TC) affects both children and adults. However, there are significant differences between them when it comes to epidemiology. Thyroid nodules in children occur rarely in comparison to adults (1.8% vs. 19–68%, respectively). Nonetheless, nodules diagnosed in children appear to be more often malignant as compared to adults (22–26% vs. 5–10%) [1]. The highest risk of malignancy refers to 15–19-year-old adolescents, especially females (in the ratio 6:1 compared to males). At that age range differentiated thyroid cancer (DTC) is the eighth most frequent cancer among males and the second most common among females [1]. The most common histological type is papillary thyroid carcinoma (PTC) constituting 80–90% of all DTC [2]. Follicular and medullary tumors are less common (9.5% and 5%, respectively) [3]. However, in spite of the high malignancy risk, children and adolescents seem to have a much better prognosis than adults (<2% mortality vs. 5.8–15%) [1,3]. Nevertheless, early and accurate diagnosis in children is extremely important.

Although several imaging methods in the clinical management of patients with thyroid nodules are available, fine needle aspiration biopsy (FNAB) currently remains the gold standard for preoperative diagnostics [1]. However, this is an invasive procedure and requires the patient’s cooperation. Therefore, attempts are still being made to develop a non-invasive tool for the management of thyroid nodules in order to limit invasive diagnostics to cases with an increased risk of malignancy. Malignant thyroid lesions are suggested by the patient’s medical history (risk factors for TC, such as genetic predisposition or prior exposure to radiation), presence of clinical symptoms (e.g., rapid nodule enlargement) or abnormal thyroid ultrasonography [4].

Thyroid ultrasonography (US) has become the most useful imaging to determine whether a thyroid nodule may require further invasive methods to reach a definitive diagnosis. The procedure is easily accessible, inexpensive and can be performed not only in specialized departments, but also in outpatient clinics. The main disadvantage of the method is the operator dependence. According to the Polish Guidelines US should always be performed in children with palpable thyroid nodules, thyroid asymmetry and/or abnormal cervical lymphadenopathy found during the physical examination. Thyroid US should also be repeated annually in every child with autoimmune thyroid disease (AITD) [5]. Many authors underline the importance of some characteristic ultrasound features of malignancy. Hypoechogenicity without a halo, shape ”taller than wide”, irregular margins, microcalcifications, chaotic vascularity, anterior subcapsular location, rapid growth and cervical lymphadenopathy are all features that indicate a malignancy in US [1,6,7,8]. However, there is no single sonographic characteristic sensitive or specific enough to identify all malignant nodules. Numerous studies have shown that the coexistence of a few suspicious features in one nodule carries a higher sensitivity and specificity for malignancy compared to each feature alone [8,9]. Therefore, various ultrasound-based risk stratification systems which combine characteristic ultrasound features of the nodules have been proposed in adults to assess the risk of malignancy [6,10,11,12]. Two commonly applied are the American Thyroid Association (ATA) ultrasound criteria for fine needle aspiration biopsy and the British Thyroid Association (BTA) ultrasound (U) classification. However, no ultrasound scale dedicated to pediatric patients has been developed to date.

In 2016, the ATA presented ultrasound criteria for FNAB in adults to introduce standard guidelines regarding the management of thyroid nodules on the basis of their ultrasound appearance and size [10]. The system classifies thyroid nodules into five categories; benign, very low suspicion, low suspicion, intermediate suspicion and high suspicion of malignancy, depending on the size and ultrasound features of a nodule (hypoechogenicity, irregular margins, presence of microcalcifications, increased intranodular blood flow and abnormal cervical lymph nodes) [10]. According to the ATA guidelines for thyroid cancer in children, the general recommendations for evaluating thyroid nodules in pediatric patients should be similar to those for adults, with the main difference being the size of the nodule [1]. While the lesion dimension in adults may be a significant factor indicating its malignancy and determining whether a given lesion should be biopsied or followed [10], in children even small lesions (<1 cm) may be malignant. Thus, it is indicated that not the size criterion, but US characteristics and clinical context should be used more preferentially to identify nodules that warrant FNAB [13]. Several classifications similar to that of the ATA are currently in use, e.g., the Thyroid Imaging Reporting and Data System (TI-RADS) with its modifications: ACR TI-RADS proposed by the American College of Radiology, EU TI-RADS recommended by the European Thyroid Association, K TI-RADS proposed by the Korean Society of Thyroid Radiology and others [11,14,15,16]. As it is published in adult patient studies, the application of the classifications may lead to a decrease in the number of unnecessary biopsies in benign nodules [10,17,18].

In 2014, the BTA developed U classification to systematize the identification of a certain stage of the thyroid nodule examined and to make it easier to determine whether or not the use of FNAB for the final diagnosis of the nodule is necessary. [6] This classification is divided into five stages (U1–U5). U1 indicates normal thyroid without nodules, U2 suggests benign lesions, whereas U3 nodules are indeterminate or equivocal and can imply a malignant nature from this group onwards. U4 nodules are suspicious for malignancy, while U5 nodules are cancerous [6,19,20]. The guidelines suggest that patients with U2 nodules without other malignancy risk factors do not require further invasive diagnosis, while patients with U3–U5 as well as U2 with increased cancer risk factors should be biopsied for cytological evaluation [6]. This classification has been proven to be a reliable screening instrument in adults for differentiating between benign and malignant thyroid lesions in several studies. Therefore, such classification makes the decision whether to perform an FNAB much easier [19,20,21,22].

In the search for complementary methods of thyroid nodule differential diagnosis, attention has been paid to the flexibility of the tissue. It has been demonstrated that most malignant tumors have an abnormal presence of collagen and myofibroblasts, which causes poor deformation of the tissue [23,24]. Elastography is a noninvasive imaging technique providing information on the stiffness of the examined area, which might help identify malignant tissues or their regions [24,25]. The result of strain elastography is presented as a strain ratio (SR) comparing the stiffness of the region of interest (ROI 1), i.e., the region of the nodule to the region of interest of the healthy tissue located at the same depth (ROI 2) as a reference. The higher the index, the harder the tissue and the greater probability that the lesion is malignant [24,26]. The cut-off for SR in the diagnosis of PTC in adults varies from 0.78 to 3.28 according to different studies [24], whereas the values for children have not been established. It is important to point out that elastography is useful generally in diagnosing papillary cancers, being the most common type in children. Other histological TC types, for example follicular cancers, have non-modified elasticity, and thus they could not be easily detected in elastography [27]. In adults, however, research shows that elastography could be useful in addition to B-mode US [27,28,29]. Studies on the use of elastography in children are rather poor and require further analysis. However, according to the Polish recommendations for the diagnosis and treatment of DTC in children, the use of elastography may help evaluate the malignancy risk in thyroid lesions in experienced and suitably equipped centers [5].

Since no ultrasound-based risk stratification system for pediatric patients has been developed, we decided to assess the application of both ATA and BTA ultrasound risk-classification systems as well as strain elastography in thyroid nodules in a group of children and adolescents and to evaluate their suitability in predicting malignant thyroid nodules.

## 2. Materials and Methods

### 2.1. Patients

A total of 165 thyroid nodules in children aged 5 to 18 years underwent thyroid ultrasonography and FNAB at the Department of Pediatric, Endocrinology, Diabetology with Cardiology Division, Medical University of Białystok. Based on the Bethesda system, 17 single thyroid nodules were selected for further analysis. All the enrolled thyroid nodules had cytology category III (atypia of undetermined significance/follicular lesion of undetermined significance, AUS/FLUS), IV (follicular neoplasm/suspicious for a follicular neoplasm, FN/SFN), V (suspicious for malignancy, SUS) or VI (malignant) according to the Bethesda System for Reporting Thyroid Cytopathology in FNAB. Written informed consent was obtained from all parents of the participating patients and children older than 16 years of age and the ethics committee approved our study. All the patients were euthyroid at the moment of biopsy.

### 2.2. Assessment of the Thyroid Hormone Concentration and Anti-Thyroid Antibody Titer

Blood for analysis was collected on an empty stomach in the morning hours from the basilic vein and centrifuged for 10 min at 2000 rpm. Sera were stored at −85 °C until the required number was collected. Serum levels of free thyroxine (fT4)and TSH were determined on electrochemiluminescence “ECLIA” with a Cobas e 411 analyzer (Roche Diagnostics, Warszawa, Poland). Normal values for fT4 ranged between 0.71 and 1.55 ng/dl and for TSH between 0.32 and 5.0 mIU/mL.Antithyroperoxidase (anti-TPO) was measured in all samples using electrochemiluminescence “ECLIA” with Modular Analytics E170 analyzer (Roche Diagnostics). The negative values for anti-TPO-Abs were between 0 and 34 IU/mL.

### 2.3. Thyroid Ultrasonography, Elastography and Fine Needle Aspiration Biopsy

Patients underwent conventional ultrasonography and ultrasound elastography followed by FNAB. One experienced ultrasonographer evaluated US features of the thyroid nodules and provided biopsy recommendations based on his own practice patterns. Both conventional US and elastography parameters were acquired with the Toshiba Aplio MX SSA-780A system equipped with a 12 MHz linear transducer. The echostructure and vascularity of the thyroid gland, as well as the nodule, were first evaluated with B-mode and Doppler US imaging. After the evaluation of ultrasonographic characteristics of the nodule, its deformation was assessed using strain elastography. Elastography was performed by a real-time free-hand technique by fivefold light compression and decompression of the thyroid tissue. The elasticity result was presented as a strain ratio (SR) which indicates the deformation of the region of interest (ROI 1), i.e., region of the nodule (avoiding cystic component and calcifications) in comparison to the region of interest of the healthy tissue (ROI 2) as a reference (ROI1/ROI2 index). This evaluation was performed automatically during the US examination using software that the ultrasound machine is equipped with. About an hour after the strain ratio assessment in every patient FNAB was obtained with ultrasonographic guidance and using the antiseptic technique. US-guided FNAB was performed with a 23-Gauge needle attached to a 5-mL disposable plastic syringe. Aspirates were spread onto glass slides and immediately fixed in 95% alcohol for both H-E staining and May-Grunwald-Giemsa staining. The criterion for an adequate smear was the presence of 6 groups of cells with >10 cells per group. Cytological diagnosis was made by a pathologist experienced in thyroid cytology and presented in the Bethesda system. If clinically indicated, the patient was operated on, and a final diagnosis was made on the basis of the histopathological result.

In a retrospective review, ultrasonographic features of each thyroid nodule were assessed according to the ATA and BTA risk stratification systems, assigning a level of malignancy risk, based on rich photographic documentation. The ATA guidelines were applied excluding the size criterion. The findings of the analysis were correlated with the available cytological/histological follow-up. A comparison was made for biopsy results and ATA guidelines, BTA U classification as well as elastography accuracy.

### 2.4. Data Analysis

All data were biostatistically processed using GraphPad Prism 8.0.0 statistical software (GraphPad Prism Inc., San Diego, CA, USA). Considering the non-normal distribution of the data within the study groups, a two-way t-Student with Mann–Whitney test was implemented. The level of statistical significance was set at the value of <0.05.

## 3. Results

Seventeen single nodules out of 165 biopsied patients (10.3%) had abnormal FNAB results and were qualified for further analysis. There were no statistically significant differences between the nodules finally diagnosed as malignant and benign in terms of age and laboratory findings, although malignant lesions were statistically significantly larger than benign lesions (Table 1).

According to the Bethesda System for Reporting Thyroid Cytopathology in FNAB two patients were assessed as stage VI (malignant), three patients as stage V (suspicious for malignancy), two patients as stage IV (follicular neoplasm/suspicious for a follicular neoplasm) and ten patients as stage III (atypia of undetermined significance/follicular lesion of undetermined significance). Histopathological results were available for ten cases after surgical treatment. The final malignant diagnosis was confirmed in histopathology in five patients out of the study group (all patients with VI and V Bethesda category). In all cases, the histological diagnosis was papillary thyroid carcinoma. Four of them were located in the right lobe (diffuse sclerosing variant with metastases to the central lymph node compartment (10/13) and lateral compartments, pT3aN1bMX, multifocal and diffused papillary carcinoma stage pT2b in both lobes withCK19+ and Ki67+ cells in immunohistochemistry, pT1b—1.6 cm in diameter—with CK19+; factor VIII+ and Ki67+ on singular cells in immunohistochemistry). The two patients with suspicion of follicular neoplasm (Bethesda IV) had no features of malignancy in histopathology. Three of the patients with AUS/FLUS had thyroid lobe removed and histologically assessed (one follicular adenoma, two others with no features of malignancy). In three other patients with AUS/FLUS, FNAB was repeated and having obtained stage II, the patients remained under close monitoring. Three other patients remained in observation (follow-up US) without being re-biopsied and one patient with this category was still diagnosed while the data were analyzed (Table 2).

Six out of seventeen analyzed patients with an abnormal cytological result (35%) had positive thyroid peroxidase antibodies (TPO) and/or lymphocytic infiltration in cytology. Two of them were malignant in the postoperative pathology.

### 3.1. Elastography

Fourteen out of seventeen patients were assessed in elastography. The highest values of the SR (over 5), indicating hard lesion, were observed in both patients with Bethesda VI (PTC in histopathology), one patient with Bethesda V (PTC In HP) and one patient with Bethesda III (benign in histopathology). The SR between 2 and 4.9, indicating intermediate lesion, was observed in one patient with Bethesda V (PTC in hp), two patients with IV as well as four patients with Bethesda III (benign in hp, benign in repeated biopsy, observed without repeating the biopsy or still being diagnosed). The SR below 2 (flexible tissue) was found in another three patients with Bethesda III (benign in repeated biopsy or observed without repeating the biopsy), indicating soft tissue. PPV for SR was 80% and NPV for SR was 100%. Figure 1 presents differences in elastography between benign and malignant thyroid nodules.

### 3.2. The ATA, BTA Risk Stratification Systems

Seventeen thyroid nodules were retrospectively evaluated according to the ATA and BTA U classifications. All cases with Bethesda VI and V which turned out to be malignant in histopathology were classified as high-risk nodules in ultrasound risk-classification systems, i.e., high level of suspicion according to ATA and U 4d, U 5a or U 5b according to BTA. Three of them were hard in elastography (SR more than 5), one was intermediate (SR 2–4.9) and one patient was not assessed in elastography. Thyroid nodules with Bethesda IV which were not confirmed on histopathological examination as malignant had intermediate stiffness in elastography (SR between 2 and 4.9) and were assessed as high level of suspicion and U 4b and U 5a, respectively. Among Bethesda III cases one nodule was hard in elastography, four were intermediate and three were soft. One hard nodule turned out to be benign in histopathology. All but one nodule with stage III corresponded with a low level of suspicion in the ATA scale and U3 in the BTA U classification.

Of the nodules with a diagnosis of PTC in histopathology (5 nodules), 100% were classified as high suspicion according to the ATA guidelines while according to the BTA U classification 80% were classified as U5 and 20% as U4. PPV for ATA was 72% and for BTA U classification 77%. Among the evaluated thyroid nodules finally classified as benign (11 nodules), 27% were classified as high suspicion and 73% as low suspicion according to the ATA guidelines. On the other hand, according to the BTA U classification, 9% of them were classified as U5, 9% as U4, 82% as U3 and 9% as U2 (Table 3). NPV for both ATA and BTA U classification was 100%.

Figure 2 and Figure 3 present malignant nodules.

Figure 4 and Figure 5 show benign nodules.

## 4. Discussion

The diagnosis of thyroid nodules is a clinical challenge for pediatric endocrinologists. As all thyroid nodules with diagnostic categories according to the Bethesda System in children are reported to be at higher malignancy risk than in adults, pediatric patients may benefit from more aggressive management [3,30]. It is particularly relevant in children with AUS/FLUS, where removal of the thyroid lobe and histopathologic examination might be recommended to make a definitive diagnosis. On the other hand, studies show that there is still a large number of thyroid procedures performed unnecessarily as no malignancy is found on histopathological examination. This is especially true for patients classified as Bethesda III or IV. Thus, a method is sought to improve the final diagnosis.

Detailed evaluation of suspicious thyroid nodules that may require surgery usually starts with ultrasonography. Over the last decade, the utility of ultrasound-based risk stratification systems in diagnostics of thyroid nodules in adults has been demonstrated. However, the usefulness of different classification systems has not yet been clearly assessed in the population of children and adolescents. Single data are available in the literature regarding the utility of assessing thyroid nodules in pediatric patients using scales such as the ATA, TI-RADS, ACR-TIRADS and EU-TIRADS, especially after exclusion of the dimension criterion [31,32,33]. Both analyzed classifications performed well in predicting malignancy in our study group, as none of the PTCs were assessed as U3, U2 or U1 in BTA nor intermediate, low or very low suspicion in ATA. The application of the ATA or BTA stratification system seems to also be useful in young patients with an indeterminate cytological diagnosis of AUS/FLUS that characterizes higher malignancy risk compared to adults, although in our study 25% of benign nodules were classified as high suspicion in ATA and 17% as U4 or U5 according to BTA U classification, indicating that the ultrasound risk stratification systems were less accurate in predicting benign nodules in children. It is worth noting that all nodules qualified as Bethesda III or IV in our study group turned out to be benign in histopathology.

In a work of Lim-Dunham et al., evaluating the diagnostic performance of US criteria of the ATA Guidelines, 39 thyroid nodules in pediatric patients were assessed. The authors indicate the utility of this method to identify nodules in children that warrant biopsy [34]. Martinez-Rios et al. retrospectively analyzed the application of two adult-based US stratification methods (ATA classification and TI-RADS) for assessment of thyroid nodules in a group of 124 children. They indicate similar characteristics of both systems to those in adults, although none of these were independently sufficient to differentiate the likelihood of malignancy [35]. Furthermore, the study of Creo et al. demonstrated similar sensitivity for detecting malignancy in thyroid nodules in children using ATA stratification and the radiologists’ overall impression, but lower specificity for detecting malignancy for ATA risk stratification. The authors highlight that none of the methods was able to precisely distinguish benign from malignant nodules, requiring FNAB for suspicious nodules [36]. The study of Richman et al. assessed the ACR TI-RADS criteria for the management of thyroid nodules in children in comparison to the ATA guidelines for the management of pediatric thyroid nodules. The researchers have shown that following only the ACR TI-RADS guidelines would reduce the biopsy rate of benign nodules, but at the same time, FNAB would not be performed in a large percentage of pediatric population cancers (22.1%). They concluded that the use of the ACR TI-RADS system in children might be inadequate [37]. Other conclusions were reached by Fernandez et al. indicating that the EU-TIRADS ultrasound criteria in children combined with clinical history are a reliable method to evaluate thyroid nodules in children. It can also be a diagnostic tool to decide which nodules are suitable for FNAB [38].

Despite a number of scientific reports on the diagnostic value of elastography in adults, there are only few studies concerning children. In the reported analysis, using real-time strain elastography, we observed that more flexible thyroid nodules are at lower risk of malignancy, as there was a statistically significantly lower SR in the group of thyroid nodules eventually diagnosed as benign in comparison to the malignant ones (3.09 vs. 6.07, *p* = 0.0356). Half of the hard lesions in our study group turned out to be malignant in postoperative pathology. Only one thyroid nodule with intermediate flexibility turned out to be malignant, whereas among soft nodules no cancers were detected in elastography. In our previous analysis of 62 thyroid nodules in pediatric patients, we demonstrated that all patients with flexible nodules showed benign cytological diagnosis, indicating a high negative predictive value of elastography for non-malignant results [39]. In the work of Cunha et al., 38 thyroid nodules in children and adolescents were assessed with ultrasound, elastography and fine-needle aspiration biopsy. Similarly, to our results high elasticity of the nodule was associated with a low risk of thyroid cancer [40].

Single reports in adults with thyroid nodules evaluating the use of both elastography together with the US stratification system are available. In a recent study by Yang et al., 205 adult patients with abnormal thyroid function were qualified to FNAB on the basis of the ACR TI-RADS. They compared strain elastography, cytology and histopathology results, showing that strain elastography for highly and moderately suspicious nodules facilitated the detection of mildly suspicious, unsuspicious and benign thyroid nodules. The authors underline the significant usefulness of strain ultrasound elastography in detecting a thyroid papillary carcinoma due to its stiffness. It is suspected, however, that other malignant tumors may have different elasticities [41]. The results of the study are consistent with the work of Hairu et al., who investigated the diagnostic efficiency of elastography and ATA guidelines in the adult population. The study revealed that elastography might be a valuable tool for the assessment of thyroid nodules qualified as highly suspicious according to ATA [42]. However, the limitation of strain elastography, the method used in our study, is the operator dependence and artifact signals from the surrounding structures such as carotid artery pulsation or tracheal motion. These artifacts can be eliminated by using shear wave elastography (SWE), which has been shown to be even more precise than strain elastography in the differential diagnosis of thyroid nodules. Luo et al. demonstrated in their work that an application of SWE in addition to ACR TI-RADS classification could improve the diagnosis of thyroid nodules. Using share wave elastography the authors identified nodules with high potential for benignity in nodules qualified as ACR TI-RADS 4. They conclude that this procedure may help identify and select benign nodules and reduce unnecessary biopsies of benign thyroid nodules [43]. The results of another study in adult patients (Yang et al.) revealed the high value of the SWE in improving the TI-RADS classification of thyroid nodules. This combination of both methods turned out to improve the sensitivity and specificity of the diagnosis of thyroid nodules. The authors indicate that SWE can be one of the non-invasive methods performed with US TI-RADS classification to support the diagnosis [44]. Similar observations were made regarding the use of SWE in combination with the TI-RADS classification in a study by Liu et al. [45] as well as in the work of Russ et al., in a large group of adult patients, showing an improvement in the prediction when TI-RADS is combined with elastography [13].

A limitation of our work is the small number of subjects. In addition, using the SWE method to assess the elasticity of thyroid nodules instead of strain elastography would eliminate the operator’s influence on the result of elastographic assessment.

## 5. Conclusions

The incidence of PTC in children and adolescents increases, but only some diagnosed thyroid nodules in a child turn out to be malignant. Thus, a reliable noninvasive method to identify which thyroid nodules require further invasive procedure is highly desirable. In our work, the BTA U classification and ATA system for risk stratification previously described for adults proved to be a suitable method for assessing the level of suspected malignancy in thyroid nodules in children. This study showed that ultrasound classification criteria are a good tool for identifying malignancy, but less effective for identifying benign nodules. Elastography as an additional diagnostic method might improve the accuracy of the differential diagnosis in children. Moreover, in multinodular goiter, elastography can be helpful in selecting which lesion should be biopsied first. If our observations are confirmed in future studies, the application of elastography together with US risk-classification systems in children may help identify most clinically significant lesions and reduce invasive procedures and unnecessary thyroidectomies in benign nodules. It is worth emphasizing that current US criteria do not replace FNAB in establishing a definitive diagnosis in thyroid nodules. Further work is needed to define thyroid nodule diagnostic methods in children, such as the US-based scoring system specific to pediatric patients, which could also include elastography.

## Figures and Tables

**Figure 1 jcm-11-01768-f001:**
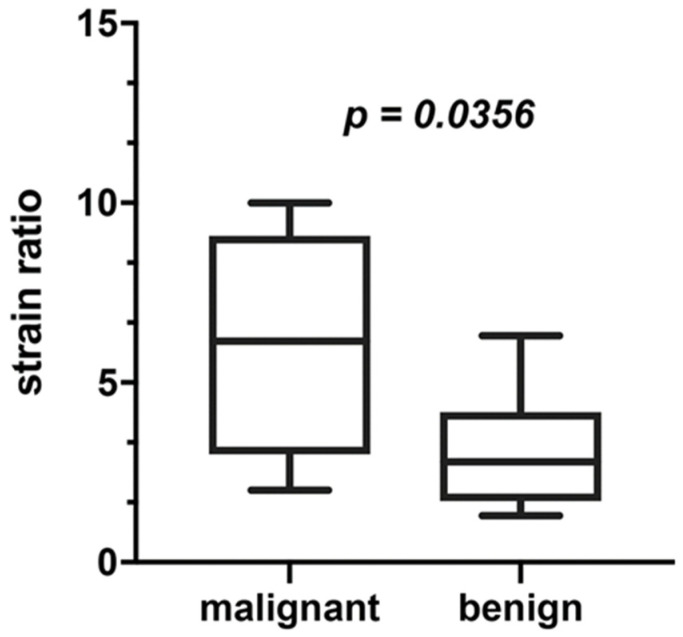
Elastography (SR) in benign and malignant thyroid nodules.

**Figure 2 jcm-11-01768-f002:**
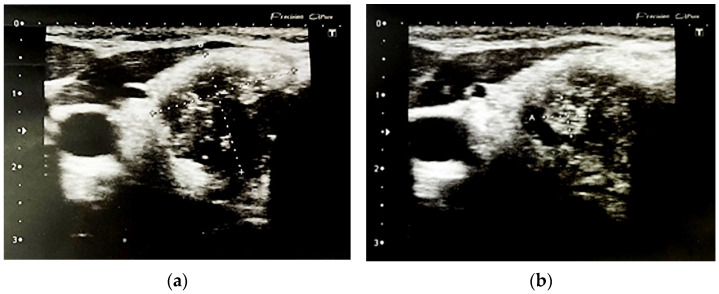
Malignant nodule in the right lobe: (**a**) solid, hyoechoic area sized 15 mm × 17 mm × 19 mm, without vascularization with numerous micro-and macrocalcifications; (**b**) with central hyperechoic area 6 mm × 4.5 mm. ATA: high suspicion, BTA: 5b.

**Figure 3 jcm-11-01768-f003:**
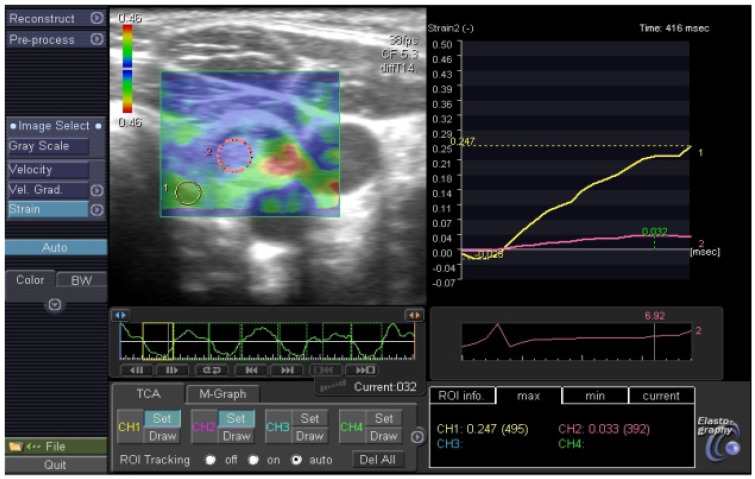
Malignant thyroid nodule in elastography, SR: 6.9.

**Figure 4 jcm-11-01768-f004:**
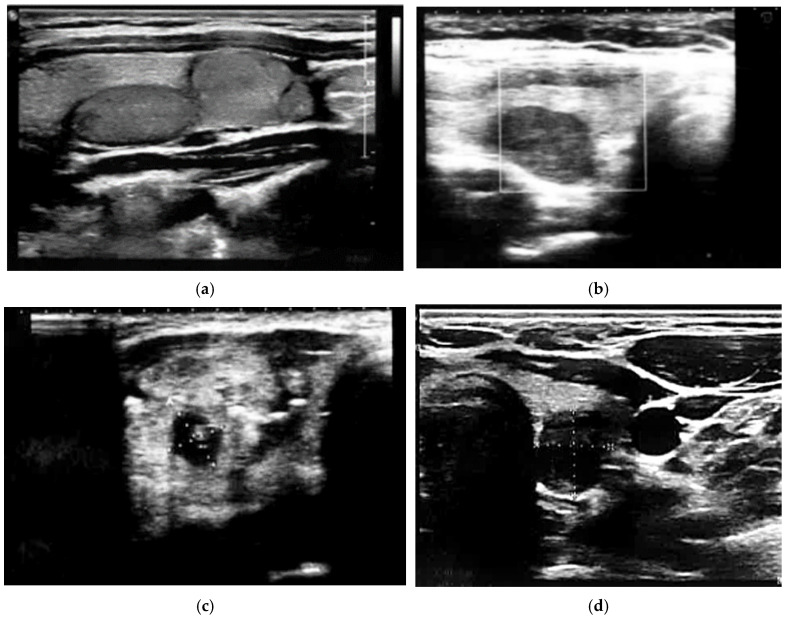
Benign thyroid nodules: (**a**) halo, isoechoic solid nodules, without microcalcifications, regular margin, ATA: low suspicion, BTA: 2a, SR 4; (**b**) hypoechoic solid nodule, without microcalcification, without microcalcification, irregular margin, ATA: low suspicion, BTA: 3b; (**c**) hyperechoic solid nodule, irregular margin, microcalcifications, ATA: high suspicion, BTA: 5a, SR: 4.7; (**d**) hypoechoic solid nodule, partially cystic, without microcalcification, ATA: low suspicion, BTA: 3b, SR: 6.3.

**Figure 5 jcm-11-01768-f005:**
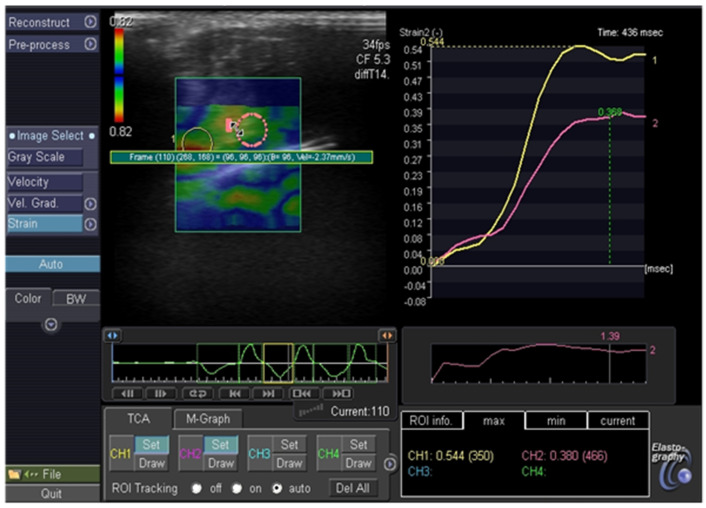
Benign thyroid nodule in elastography, SR: 1.39.

**Table 1 jcm-11-01768-t001:** Characteristics of the study group.

	All (Mean ± SD)	Malignant (Mean ± SD)	Benign (Mean ± SD)	*p*
number of patients	17 ^a^	5	11	
sex (boys/girls)	4/13	1/4	3/9	
age (years)	5–18 (15.29 ± 3.27)	14–18 (16.4 ± 1.57)	5–18 (14.8 ± 3.7)	ns
history of cervical irradiation	0/17	0/5	0/11	
nodular goitre in family history	4/17	1/5	3/11	
palpable thyroid nodule	3/17	1/5	2/11	
TPO (μlU/L)	1–367 (92.7 ± 122.3)	9.4–243 (114.8 ± 118.5)	1–367 (85.4 ± 129.6)	ns
size I (mm)	3.4–21.0 (12.22 ± 5.6)	6.0–21.0 (15.3 ± 6.16)	3.4–17.9 (10.9 ± 5.13)	<0.01
size II (mm)	2.0–22.6 (10.8 ± 6.6)	4.5–19.0 (13.1 ± 5.86)	2.0–22.6 (9.9 ± 6.95)	<0.05

ns—not statistically significant, ^a^ one patient was still undergoing invasive diagnostics at the time of writing the article.

**Table 2 jcm-11-01768-t002:** The results of FNAB, histopathology, elastography, ATA and BTA risk stratification systems.

Patient	Sex	Lobe	Nodule Size (mm × mm)	SR	FNAB (Bethesda)	ATA Classification	BTA U Classification	Histopathology
1	F	R	21 × 15	6.3	VI	high suspicion	U 5b	PTC
2	M	R	20 × 17	10	VI	high suspicion	U 5b	PTC
3	F	R	6 × 4.5	2	V	high suspicion	U 4d	PTC
4	F	R	19 × 17	nm	V	high suspicion	U 5a	PTC
5	F	L	12.5 × 10	6	V	high suspicion	U 5b	PTC
6	F	R	7 × 5.7	4.7	IV	high suspicion	U 5a	benign
7	F	R	16 × 16	3.6	IV	high suspicion	U 4b	benign
8	F	R	17.9 × 7.4	4	III	low suspicion	U 2a	benign
9	M	R	6.5 × 4	1.7	III	low suspicion	U 3b	clinical observation
10	M	L	22.6 × 15	2.6	III	low suspicion	U 3c	benign in repeated FNAB
11	F	R	13 × 9.5	nm	III	low suspicion	U 3b	follicular adenoma
12	F	L	7 × 6	nm	III	low suspicion	U 3c	benign in repeated FNAB
13	F	L	12 × 10	6.3	III	low suspicion	U 3b	benign
14	M	R	3.4 × 2	3	III	low suspicion	U 3b	clinical observation
15	F	R	6.8 × 5	1.7	III	high suspicion	U 3c	clinical observation
16	F	R	5.6 × 4.5	1.3	III	low suspicion	U 3c	benign in repeated FNAB
17	F	R	22 × 15	2	III	low suspicion	U 3b	at diagnosis

F—female, M—male, R—right lobe, L—left lobe, SR—strain ratio, nm—not measured, PTC—papillary thyroid cancer.

**Table 3 jcm-11-01768-t003:** Ultrasound risk-classification system results in benign and malignant thyroid nodules.

		Final Diagnosis Malignant	Final Diagnosis Benign	Sensitivity	Specificity
ATA classification	high suspicion	100%	27%	100.00% 95% CI: 47.82–100.00%	75.00% 95% CI: 42.81–94.51%
	intermediate suspicion	-	-		
	low suspicion	-	73%		
	very low suspicion	-	-		
	benign	-	-		
BTA U classification	U5	80%	9%	80.00% 95% CI: 28.36–99.49%	91.67% 95% CI: 61.52–99.79%
	U4	20%	9%		
	U3	-	83%		
	U2	-	9%		
	U1	-	-		

## Data Availability

Data Availability Statements in section University Children Hospital in Bialystok, Poland.

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
