# Peer review of "Suspected Malignant Thyroid Nodules in Children and Adolescents According to Ultrasound Elastography and Ultrasound-Based Risk Stratification Systems—Experience from One Center"

_jcm, 2022, doi:10.3390/jcm11071768_

Round 1

Reviewer 1 Report

In the present paper, the authors report an Institutional experience over the application of the ATA and BTA ultrasound risk-classification systems and strain elastography in the management of thyroid nodules in children and adolescents, suggesting that application of these stratification system and elastography may improve differential diagnosis and help the physician to decide about the need for further invasive diagnosis of thyroid nodules in children. They report on a case series of 165 thyroid nodules in children aged 5 to 18 years underwent thyroid ultrasonography and FNAB, seventeen of which underwent.

To my opinion, the paper is well written, minor changes should be edited. In order to gain a wider experience we suggest the authors to conceive a multicentric study thus improving reliability of the presented methodology, especially when considering pediatric age (cf. Thyroid cancer in adolescents and young adults. Massimino M et al Pediatr Blood Cancer. 2018) (cf. Factors associated with postoperative hypocalcemia following thyroidectomy in childhood. Spinelli C et al. Pediatr Blood Cancer. 2022).

Author Response

Responses to Reviewers’ Comments

Manuscript ID: jcm-1636435

Title: Suspected malignant thyroid nodules in children and adolescents according to Ultrasound Elastography and Ultrasound-Based Risk Stratification Systems – experience from one center.

Authors: Hanna Borysewicz-Sańczyk, Beata Sawicka, Agata Karny, Filip Bossowski, Katarzyna Marcinkiewicz, Aleksandra Rusak, Janusz Dzięcioł, Artur Bossowski

Submitted to section: Endocrinology & Metabolism

To my opinion, the paper is well written, minor changes should be edited. In order to gain a wider experience we suggest the authors to conceive a multicentric study thus improving reliability of the presented methodology, especially when considering pediatric age (cf. Thyroid cancer in adolescents and young adults. Massimino M et al Pediatr Blood Cancer. 2018) (cf. Factors associated with postoperative hypocalcemia following thyroidectomy in childhood. Spinelli C et al. Pediatr Blood Cancer. 2022).

Thank you very much for your valuable suggestions. To gain a wider experience we are planning to expand our study to other centers in Poland and abroad.

Reviewer 2 Report

The Conclusion needs to be revised. 

It should be specific and relevant to the study observations 

which is missing in the conclusion currently.

The small number of patients is a limitation. There is already another published paediatric study available with higher patient numbers. What additional information is obtained in this study?

Author Response

Responses to Reviewers’ Comments

Manuscript ID: jcm-1636435

Title: Suspected malignant thyroid nodules in children and adoles-cents according to Ultrasound Elastography and Ultra-sound-Based Risk Stratification Systems – experience from one center.

Authors: Hanna Borysewicz-Sańczyk, Beata Sawicka, Agata Karny, Filip Bossowski, Katarzyna Marcinkiewicz, Aleksandra Rusak, Janusz Dzięcioł, Artur Bossowski

Submitted to section: Endocrinology & Metabolism

The Conclusion needs to be revised. It should be specific and relevant to the study observations which is missing in the conclusion currently.

Thank you. The conclusions have been supplement.

The small number of patients is a limitation. There is already another published paediatric study available with higher patient numbers.

Thank you for that remark. Indeed, the size of the study group is small, thus we added this information as a limitation to the manuscript. We are planning to expand our study to other centers in Poland and abroad.

What additional information is obtained in this study?

Thank you. The additional information obtained in this study is that in cases of multinodular goiter, elastography can be helpful in selecting which lesion should be biopsied firstly. This information was added to the conclusions.

This manuscript is a resubmission of an earlier submission. The following is a list of the peer review reports and author responses from that submission.

Round 1

Reviewer 1 Report

  1. You have a biased and small sample
  2. Includes patient with only one FNAB
  3. Does not include benign and operated patients
  4. it is impossible to show that elastography is useful in such a small sample
  5. I suggest the authors complete at least 30 cases per group, with histopathology and again analyze the issue.

Author Response

Responses to Reviewers’ Comments

Manuscript ID: jcm-1476237

Title: The application of ultrasound elastography in combination with ultrasound-based risk stratification systems in diagnosis of malignant thyroid nodules in children and adolescents – retrospective analysis of one center

Authors: Hanna Borysewicz-Sańczyk, Beata Sawicka, Agata Karny, Filip Bossowski, Katarzyna Marcinkiewicz, Aleksandra Rusak, Janusz Dzięcioł, Artur Bossowski

Submitted to section: Endocrinology & Metabolism

  1. You have a biased and small sample

Thank you for that remark. Indeed, the size of the study group is small, thus we added this information as a limitation to the manuscript.

  1. Includes patient with only one FNAB

Thank you. According to the Polish guidelines [5] among cytopathological findings categorised according to the Bethesda System for Reporting Thyroid Cytopathology (Bethesda) as Bethesda III and IV in children, thyroid cancer is diagnosed postoperatively more often than in adults, thus surgery (lobectomy with isthmectomy) should be recommended in these groups instead of repeated FNAB. However in three patients with Bethesda III FNAB was repeated (see Table 2). Patients with Bethesda V and VI had surgery without repeating FNAB.

  1. Does not include benign and operated patients

Thank you for that remark. In children with benign nodules (Bethesda II) the risk for malignancy is estimated for about 5% and the routine surgical treatment is not recommended. However, if accompanied by clinical or ultrasonographic features of malignancy they might require surgery, but we had no such patients in our group of 165 children.

  1. it is impossible to show that elastography is useful in such a small sample

Thank you for that suggestion. Between 2013 and 2020 we examined nearly 300 thyroid nodules in children (data not shown). We observed that nodules with strain ratio 3 and above were at a higher risk of malignancy. For the present study 165 patients were classified according to the subject of the study.

  1. I suggest the authors complete at least 30 cases per group, with histopathology and again analyze the issue.

Thank you for that remark. This is a valuable suggestion. We are going to continue our study in our department in further investigations to enlarge the study group. As the incidence of thyroid nodules in children is not as high as in adults, it is impossible to gather about 30 new cases in such a short time.

Reviewer 2 Report

The study is globally well written and well documented.

The aim of the study is well-defined.

The methodology and the statistical analysis should be reviewed/reconsidered.

Major revision

Abstract

English language should be improve

“with presence”: with THE presence

In adults ultrasonographic: add a coma between adults and ultrasonographic

“Furthermore elastography”: add a coma

Authors should avoid abbreviation in the abstract: ATA and BTA should be completely written

“with Bethesda III or higher”: it means authors have included BETHESDA IV, V and VI ?, please clarify in the abstract section this point

was statistically significant higher: authors should write “significantly higher” (and remove statistically)

“According to ATA guidelines 100%”: add a coma after guidelines

The methods section is not clear in the abstract

Authors should be attentive to the punctuation +++

What are U4 or U3: authors should be more specific

Introduction

The introduction is well-written and well-documented.

Methods

Authors should describe in this section, the biological data:

-“Biological: thyroid function tests (TSH, FT4, FT3, thyroid peroxidase (TPO) antibodies)”

-For example: “Laboratory tests were performed in the hospital lab with routine assay kits: TSH, anti-TPO and anti-thyroglobulin antibodies were measured with, respectively, UniCell® DxI 800  immunoassay System (Beckman Coulter, Inc) using Access TSH 3rd IS (normal range [0.4–3.6 μIU/mL]), Access TPO antibodies”

Did the authors performed calcitonin ?

Table 1 is clear and interesting: authors should add calcitonin

Maybe this table 1 should be positioned in the “results section”, as it already give us some interesting results concerning the whole cohort

Authors should clarify if some young patients had previous cervical irradiation or family history of thyroid diseases (in the table 1)

I am a bit surprising that the authors have tried to conclude about elastography, but unfortunately they did not use the more precise and efficient method for that (share wave elastography (SWE) >>>>>> Strain elastography); it constitutes an significant limitations of the present study

“About an hour after the strain ratio assessment in every patient FNAB was obtained”: the authors should develop and clarify the way the FNAB was performed (how many needles? Size of the needles Gauge? the conventional method, where cells are layered on a glass slide, immediately after their extraction ? and Liquid Based Cytology (LBC) ?

Results

Please clarify if the patients have some cervical complaint? Or palpable lymph nodes or other symptoms (table 1.)

“Final malignant diagnosis was confirmed in histopathology in five patients out of the study group (all patients with VI and V Bethesda category)”: the authors should very carefully detailed the characteristics of these thyroid cancer (Multifocality? size of the tumor? Extension? Lymph nodes? BRAF mutation? And the pTNM classification)

Tables 2 and 3: clear

Figure 1: clear

Legends are clear and accurate

Table 3: authors should try to add sensitivity and specificity value of ATA and BTA score

All the iconography is very clear and comprehensive. Very good point.

Statistics: why the authors should not try to pool ATA or BTA and SE together and correlate them with cytological and histological results (as a global/composite endpoint)? Not enough nodules?

Discussion/conclusion

Authors should add the limitations of their study: obviously, the small size of the cohort

Author Response

Responses to Reviewers’ Comments

Manuscript ID: jcm-1476237

Title: The application of ultrasound elastography in combination with ultrasound-based risk stratification systems in diagnosis of malignant thyroid nodules in children and adolescents – retrospective analysis of one center

Authors: Hanna Borysewicz-Sańczyk, Beata Sawicka, Agata Karny, Filip Bossowski, Katarzyna Marcinkiewicz, Aleksandra Rusak, Janusz Dzięcioł, Artur Bossowski

Submitted to section: Endocrinology & Metabolism

Major revision

Abstract

English language should be improve

“with presence”: with THE presence

In adults ultrasonographic: add a coma between adults and ultrasonographic

“Furthermore elastography”: add a coma

Thank you. English was checked and improved.

Authors should avoid abbreviation in the abstract: ATA and BTA should be completely written

Agree. The abbreviations were explained.

“with Bethesda III or higher”: it means authors have included BETHESDA IV, V and VI ?, please clarify in the abstract section this point

Thank you. It was clarified.

was statistically significant higher: authors should write “significantly higher” (and remove statistically)

Corrected.

“According to ATA guidelines 100%”: add a coma after guidelines

It was corrected.

The methods section is not clear in the abstract

Thank you. This section was described.

Authors should be attentive to the punctuation +++

Thank you. It was corrected.

What are U4 or U3: authors should be more specific

Agree. The abbreviations were explained.

Introduction

The introduction is well-written and well-documented.

Methods

Authors should describe in this section, the biological data:

-“Biological: thyroid function tests (TSH, FT4, FT3, thyroid peroxidase (TPO) antibodies)”

-For example: “Laboratory tests were performed in the hospital lab with routine assay kits: TSH, anti-TPO and anti-thyroglobulin antibodies were measured with, respectively, UniCell® DxI 800 immunoassay System (Beckman Coulter, Inc) using Access TSH 3rd IS (normal range [0.4–3.6 μIU/mL]), Access TPO antibodies”

Thank you for that remark. The biological data were added to this section.

Did the authors performed calcitonin ?

This is a valuable suggestion. We did not routinely asses calcitonin concentraion in our patients because it is not routinely recomended in the diagnosis of thyroid nodules in children in the absence of a family history of MEN, medullary cancer or RET mutations.

Table 1 is clear and interesting: authors should add calcitonin

Thank you. We did not asses calcitonin concentraion.

Maybe this table 1 should be positioned in the “results section”, as it already give us some interesting results concerning the whole cohort

Thank you for that remark. Table 1 has been moved to “results section”.

Authors should clarify if some young patients had previous cervical irradiation or family history of thyroid diseases (in the table 1)

Thank you for this suggestion. Medical documentation and data regarding previous exposure to cervical irradiation, as well as family history of thyroid disease, were verified. The data are included in Table 1.

I am a bit surprising that the authors have tried to conclude about elastography, but unfortunately they did not use the more precise and efficient method for that (share wave elastography (SWE) >>>>>> Strain elastography); it constitutes an significant limitations of the present study

Thank you. This is a valuable suggestion for the future investigations. Between 2013 and 2020 we examined nearly 300 thyroid nodules in children with Toshiba Aplio MX SSA-780A system, analyzing strain ratio (165 patients were classified to the present work according to the subject of the study). We currently have the ability to assess SWE but this was not possible in the patients included in the study.

“About an hour after the strain ratio assessment in every patient FNAB was obtained”: the authors should develop and clarify the way the FNAB was performed (how many needles? Size of the needles Gauge? the conventional method, where cells are layered on a glass slide, immediately after their extraction ? and Liquid Based Cytology (LBC) ?

Thank you. This is a valuable suggestion. The FNAB description was added to “Materials and methods” section.

Results

Please clarify if the patients have some cervical complaint? Or palpable lymph nodes or other symptoms (table 1.)

Thank you, this is a very helpful suggestion. Medical documentation and data regarding symptoms were verified. The data are included in Table 1.

“Final malignant diagnosis was confirmed in histopathology in five patients out of the study group (all patients with VI and V Bethesda category)”: the authors should very carefully detailed the characteristics of these thyroid cancer (Multifocality? size of the tumor? Extension? Lymph nodes? BRAF mutation? And the pTNM classification)

Thank you. This is a valuable suggestion. The data available in the medical documentation were included. According to the literature it is possible to study some molecular markers of thyroid cancers in children (e.g.BRAF V600E mutation, RET fusions, or TERT mutation). We did not perform such tests in the study group.

Tables 2 and 3: clear

Figure 1: clear

Legends are clear and accurate

Table 3: authors should try to add sensitivity and specificity value of ATA and BTA score

The statistical analysis was checked, the data are enclosed in Table 3.

All the iconography is very clear and comprehensive. Very good point.

Statistics: why the authors should not try to pool ATA or BTA and SE together and correlate them with cytological and histological results (as a global/composite endpoint)? Not enough nodules?

Thank you for this suggestion. Combitation of ATA/BTA with SE and correlation with cytological and histological results would not give reliable results due to insufficient number of nodules.

Discussion/conclusion

Authors should add the limitations of their study: obviously, the small size of the cohort

The limitations of the study was added into the manuscript.

Reviewer 3 Report

This seems to be a pre-final version of the manuscript, so my evaluation is all in all limited.

Generally, the topic is relevant for the pediatrician, as fine needle biopsy might indeed be stressful for child and interventionalist.

However, the work can not satisfy expectations induced by the title: Although ATA and BTA classification and (in most cases) strain elastography was performed, the authors do not work out, how strain elastography can improve specificity and sensitivity of these classification systems. This could be a very interesting point, which was failed to assess.

Moreover, authors should provide positive predictive value and negative predicitve value, which a much more intuitive. Table 3 is hard to understand, CIs are interesting. More individuals should be included.

Line 350: probably typing error "p= 0.36".

All in all, the manuscript needs distinct improvement.

Author Response

Responses to Reviewers’ Comments

Manuscript ID: jcm-1476237

Title: The application of ultrasound elastography in combination with ultrasound-based risk stratification systems in diagnosis of malignant thyroid nodules in children and adolescents – retrospective analysis of one center

Authors: Hanna Borysewicz-Sańczyk, Beata Sawicka, Agata Karny, Filip Bossowski, Katarzyna Marcinkiewicz, Aleksandra Rusak, Janusz Dzięcioł, Artur Bossowski

Submitted to section: Endocrinology & Metabolism

Comments and Suggestions for Authors

This seems to be a pre-final version of the manuscript, so my evaluation is all in all limited.

Generally, the topic is relevant for the pediatrician, as fine needle biopsy might indeed be stressful for child and interventionalist.

However, the work can not satisfy expectations induced by the title: Although ATA and BTA classification and (in most cases) strain elastography was performed, the authors do not work out, how strain elastography can improve specificity and sensitivity of these classification systems. This could be a very interesting point, which was failed to assess.

Thank you. W proposed a new title “Retrospective assessment of malignant thyroid nodules in a group of children and adolescents according to BTA U classification and ACR TI-RADS ultrasound-based risk stratification system in combination with elastography.”

Moreover, authors should provide positive predictive value and negative predicitve value, which a much more intuitive.

Thank you. Positive predictive value and negative predicitve value were added.

Strain ratio  PPV = 80%, NPV = 100%

ATA  PPV = 72%, NPV = 100%

BTA  PPV = 77%, NPV = 100%

Table 3 is hard to understand, CIs are interesting.

Thank you. Table 3 has been improved.

More individuals should be included.

Thank you for that remark. This is a valuable suggestion. We are going to continue our study in our department in further investigations to enlarge the study group. As the incidence of thyroid nodules in children is not as high as in adults, it is impossible to gather about 30 new cases in such a short time. The small number of subjects in our study was taken into account as a limitation of our work.

Line 350: probably typing error "p= 0.36".

Thank you. It was corrected.

Round 2

Reviewer 1 Report

Again, This work cannot be published because it has a design and a sample that is too small for its results to be real.

Author Response

None.

Reviewer 2 Report

The answer to the reviewer is fine.

No additional comment.

Author Response

None.

Reviewer 3 Report

As criticised initally, the title suggests that different classification systems and strain elastography in combination were used to perform risk stratification, which is not the case (although it could be with proper analysis). The suggested new title (instead adapting the analysis) does not help to avoid this assumption.

All in all the manuscript has several weaknesses, mostly regarding the design of the study. Why were only 17 patients included and not all 165? Including patients with certain histologic criteria only distorts the estimation of specificity and sensitivity of the classification systems used (selection bias).